# How Should We Assign Large Infiltrative Hepatocellular Carcinomas for Staging?

**DOI:** 10.3390/cancers12092589

**Published:** 2020-09-10

**Authors:** Yoo Jin Lee, Yoo Ra Lee, Chung Gyo Seo, Hyun Gil Goh, Tae Hyung Kim, Sun Young Yim, Na Yeon Han, Jae Min Lee, Hyuk Soon Choi, Eun Sun Kim, Bora Keum, Hyonggin An, Beomjin Park, Yeon Seok Seo, Hyung Joon Yim, Ji Hoon Kim, Young Dong Yu, Dong Sik Kim, Yoon Tae Jeen, Hoon Jai Chun, Hong Sik Lee, Chang Duck Kim, Soon Ho Um

**Affiliations:** 1Department of Pathology, Korea University Medical Center, Seoul 136-701, Korea; yujinn87@korea.ac.kr (Y.J.L.); silverkes@korea.ac.kr (E.S.K.); 2Division of Gastroenterology and Hepatology, Department of Internal Medicine, Korea University Medical Center, Seoul 136-701, Korea; lyr427@korea.ac.kr (Y.R.L.); seoprides@korea.ac.kr (C.G.S.); goghg@korea.ac.kr (H.G.G.); lacid@korea.ac.kr (T.H.K.); jmlee80@korea.ac.kr (J.M.L.); mik77@korea.ac.kr (H.S.C.); borakeum@korea.ac.kr (B.K.); drseo@korea.ac.kr (Y.S.S.); gudwns21@korea.ac.kr (H.J.Y.); kjhhepar@korea.ac.kr (J.H.K.); ytjeen@korea.ac.kr (Y.T.J.); drchunhj@korea.ac.kr (H.J.C.); hslee60@korea.ac.kr (H.S.L.); kumcge@korea.ac.kr (C.D.K.); 3Department of Radiology, Korea University Medical Center, Seoul 136-701, Korea; mammos2000@korea.ac.kr (N.Y.H.); rupture226@korea.ac.kr (B.P.); 4Department of Biostatistics, Korea University Medical Center, Seoul 136-701, Korea; hyonggin@korea.ac.kr; 5Department of Surgery, Korea University Medical Center, Seoul 136-701, Korea; hust1351@korea.ac.kr (Y.D.Y.); kimds1@korea.ac.kr (D.S.K.)

**Keywords:** hepatocellular carcinoma, AJCC 8th staging system, gross morphology, infiltrative type, nodular type, overall survival, prognostic efficacy

## Abstract

**Simple Summary:**

The outcome of hepatocellular carcinoma (HCC) patient varies depending on tumor burden and liver function. The BCLC staging system is based on these two factors while AJCC staging system is based on tumor burden only. However, the outcome of HCC does not solely depend on these two factors, but also the aggressiveness of tumor behavior represented by tumor morphology. The morphology of HCC can be divided into three types; nodular, multinodular confluent and infiltrative type. The infiltrative type is known to be associated with poor prognosis and should be staged differently from other tumors. This study revealed that large infiltrative type HCC (≥4 cm) was associated with worse survival especially in early AJCC T-stages (T1b/2) and BCLC stages (A/B). In addition, reassignment of large infiltrative tumor to T3 and T4 and to BCLC B and C increased the discriminatory ability of each staging system.

**Abstract:**

Infiltrative gross morphology of hepatocellular carcinoma (HCC) is known to be associated with poor prognosis, but this is not considered for staging. A total of 774 HCC patients who underwent curative liver resection were retrospectively reviewed and the prognostic significance of infiltrative type HCC was assessed using the American Joint Committee on Cancer (AJCC) and Barcelona Clinic Liver Cancer (BCLC) staging systems. Seventy-four patients (9.6%) had infiltrative HCCs with a higher proportion of multifocal tumors, larger tumors, vessel invasion, increased tumor marker levels, and advanced T-stages than those with nodular HCC (all, *p* < 0.01). Infiltrative morphology was independently associated with lower overall survival (OS), but its impact was significant when the tumor size was ≥ 4 cm (*p* < 0.001). Under current AJCC and BCLC staging criteria, these large infiltrative HCCs were associated with significantly worse OS in early AJCC T-stages (T1b/T2, *p* < 0.001) and BCLC stage A/B (both, *p* < 0.01) but not in late AJCC (T3/T4) and BCLC C. The reassignment of this subtype to T3 and T4 increased the discriminatory ability of AJCC T-staging with lower AIC values (3090 and 3088 vs. 3109) and higher c-index (0.69 and 0.69 vs. 0.67), respectively (both, *p* < 0.001). Similarly, the reassignment of large infiltrative HCC to BCLC stages B and C also improved the prognostic performance. Large infiltrative HCCs should be assigned to more advanced stages in current staging systems for their prognostic impact.

## 1. Introduction

Hepatocellular carcinoma (HCC) is the sixth most common cancer and is the second most lethal cancer type globally [1,2]. Despite the overall poor prognostic nature of HCC, the patient outcome varies depending on the tumor burden and liver function in each patient [3]. Tumor burden constitutes the basis of tumor staging that helps to assess the prognosis of cancer patients and to plan their management. The American Joint Committee on Cancer (AJCC) system for HCC is a standard tumor staging system dealing with tumor burden, defined by the size, number, and vessel invasion of tumors in the liver, and their nodal and systemic metastasis. The AJCC staging system was developed and validated using patients who underwent liver resection [4,5]. Meanwhile, the unique role of liver function when assessing the prognosis of HCC patients led to the development of the so-called “clinical staging systems for HCC,” which incorporates the markers of liver function as well as the tumor extent. The Barcelona Clinic Liver Cancer (BCLC) staging system is a well-known clinical staging scheme for HCC patients that properly stratifies patients according to prognosis, simultaneously linking up with treatment indication [6].

However, a considerable portion of prognostic heterogeneity still remains even when employing these staging systems, as the outcome of HCC does not solely depend on tumor burden but also varies with the aggressiveness of tumor behavior. In particular, HCCs exhibiting infiltrative patterns on gross morphology are associated with a poor survival of patients [7,8,9]. With this perspective, we hypothesized that the gross morphologic features of HCC can help in improving the prognostic performance of staging systems as they reflect the pattern of growth that represents tumor behavior.

The traditional morphologic classification of Eggel divides HCCs into three major forms. The nodular form occurs as solitary or multiple nodules with variable sizes that are sharply demarcated, while the massive form appears as a large infiltrative mass that is poorly demarcated, usually occupying at least a whole segment of the liver and frequently accompanied by small intrahepatic metastatic foci. The diffuse form, although very rare, involves nearly the whole liver and lacks a well-demarcated boundary and blends into the background of the cirrhotic liver [10]. The Liver Cancer Study Group of Japan also classified the macroscopic morphologic features of HCC in a similar way: nodular (expanding nodular, nodular with perinodular extension), multinodular confluent, and infiltrative types [11]. The HCCs showing infiltrative morphology are commonly associated with a higher rate of microvascular invasion (MVI) and a higher level of serum alpha-fetoprotein (AFP) with worse prognosis than nodular or multinodular confluent HCCs in the same stage [9,12]. Consequently, this subtype is often present in advanced disease, and frequently spreads through the liver via the tumor thrombus of the portal vein [7]. The large infiltrative HCCs in Japanese criteria share morphologic features with the massive form of the Eggel classification.

Despite such differing tumor behaviors according to gross morphology, current staging systems handle the infiltrative type HCC the same as the nodular type, only taking into consideration the size, number, and presence of vessel invasion. In addition, the spreading and permeative appearance of large infiltrative HCCs often hinders accurate counting of their numbers on computed tomography (CT) or magnetic resonance imaging (MRI). This precludes adequate staging of the large infiltrative type HCC, unlike the nodular type.

To date, there has been no study that fully addressed the potential role of tumor morphology in staging HCC, although it requires further clarification. Therefore, we analyzed the prognostic impact of the infiltrative type HCC by evaluating patients who underwent liver resection for HCC, and attempted to clarify how to assign this HCC subtype in the current staging systems to increase their discriminatory ability.

## 2. Methods

### 2.1. Patient Selection

From January 2004 to December 2016, 992 consecutive patients who underwent liver resection on the imaging diagnosis of HCC in three tertiary-care hospitals affiliated to Korea University (Anam, Ansan and Guro) were retrospectively investigated. Among them, 774 patients were finally included in the statistical analysis, excluding those with masses diagnosed with HCC on imaging but not confirmed on the resected specimen (dysplastic nodule, intrahepatic cholangiocarcinoma, mixed hepatocellular-cholangiocarcinoma, and secondary metastatic tumor) (*n* = 92), those who underwent locoregional therapies such as radiofrequency ablation (RFA) or transarterial chemoembolization (TACE) more than once before resection (*n* = 64), those who were lost to follow-up within 6 months of surgery (*n* = 35), those with residual tumors after resection (*n* = 16), and those with perioperative death (*n* = 11). The flow chart of patient selection is shown in Figure 1. The study was approved by the institutional review board of Korea University (2017AN0199) and the need for informed consent was waived.

### 2.2. Clinical Findings

Clinical data for each patient included age, gender, laboratory results (complete blood count, liver battery, α-fetoprotein (AFP), protein induced by vitamin K absence or antagonist-II (PIVKA-II)), and etiologies of liver disease. Chronic hepatitis B patients were administered antiviral treatment if indicated. The mortality was assessed at the end of the maximum follow-up period based on medical records and the Korea National Health Insurance Service.

### 2.3. Imaging Findings

The preoperative diagnosis of HCC was based on imaging criteria using CT and MRI with contrast agents. According to the EASL or AASLD guidelines, nodules detected before 2011 were diagnosed as HCC if they had typical appearance of HCC in two and one imaging studies for nodules between 1 and 2 cm and larger than 2 cm, respectively [13,14]. However, after 2011, one dynamic imaging study with typical findings was sufficient to diagnose HCC for nodules larger than 1 cm [15].

Nodular HCC was defined as tumors with well-circumscribed borders showing typical radiologic characteristics of HCC, with early arterial hyperenhancement and washout patterns on portal venous phase images. Infiltrative HCC was defined when tumors presented poorly demarcated indistinct margins with diffuse permeative appearance, demonstrating very inhomogeneous or miliary enhancement patterns on arterial phase images and corresponding washout on more delayed phase images (Appendix A).

### 2.4. Pathologic Findings

Surgical resection was performed under general anesthesia using standard hepatectomy techniques by experienced surgeons (K.D.S. and Y.Y.D. et al.). Pathologic data from the resected specimens included tumor size, tumor number, and the presence of microvascular invasion, gross morphology including nodular, multinodular confluent, and infiltrative types classified according to the General Rules for the Study of Primary Liver Cancer published by the Korean Liver Cancer Association, were analyzed by experienced pathologists (K.J.Y. and L.Y.J. et al.) [16].

The infiltrative type HCC is defined as a mass with foci varying in size, which fuse to form a larger foci without a distinct margin, or a mass with a permeative appearance, which blends into the background of the cirrhotic liver with an indistinct margin [11]. The most representative findings of the nodular and infiltrative type HCCs are shown in Appendix A.

### 2.5. Staging System

The AJCC staging was based on the 8th version using pathologic data on the size and number of tumors, the presence of vascular invasion, and lymph node involvement [17]. Microvascular invasion was considered when classifying patients into T1a, T2, and T3 stages, while patients were classified as T4 when there was a gross invasion of major vessels such as the portal vein and hepatic vein. The BCLC staging was based on imaging findings and only the gross invasion of major vessels was considered in the staging, not the microvascular invasion [6]. In adherence to the original BCLC staging system, any single tumor without gross vascular invasion was classified as BCLC A, disregarding the tumor size. In addition to tumor burden, liver functional reserve and patient performance status was determined by the Child–Pugh classification and Eastern Cooperative Oncology Group, respectively [6].

### 2.6. Statistical Analysis

Continuous and categorical variables of the baseline clinical characteristics were compared using the Mann–Whitney *U* test and χ^2^ test, respectively. The cumulative incidence of OS was determined according to the AJCC T-stage and BCLC staging system and the impact of the infiltrative type HCC on each staging system was evaluated using Kaplan–Meier plots (log-rank test), censoring the patients who were lost to follow-up.

The prognostic power of individual staging systems was assessed based on the homogeneity (survival among the subjects in the same stage within each system) and discriminatory ability (survival difference among different stages within each system). The homogeneity of each system was determined by the likelihood ratio (LHR) χ^2^ test based on a Cox proportional hazard regression model. We also presented the results of Cox’s regression using the Akaike information criterion (AIC) by counting stages within each system as categorical variables; the smaller AIC value indicated the better performance of a model [18]. The discriminatory ability of each staging system was evaluated using Harrell’s concordance index (c-index). Comparison of the c-indices among the staging systems was performed with rcorrp.cens in the “Hmisc” package in R [19]. Bootstraps with 1000 resample were used for these activities. All statistical analyses were performed using SPSS software (SPSS version 20.1; SPSS, Chicago, IL, USA) and R version 3.4.3 (The R project, Vienna, Austria).

## 3. Results

### 3.1. Baseline Clinical and Tumor Characteristics

Demographic and clinicopathologic data of 774 patients at baseline are shown in Table 1. Overall, 40.3% of the patients were older than 60 years, 78.4% were male, and 77.6% were infected with the hepatitis B virus (HBV). Based on the pathologic findings, tumors were unifocal in most patients (78.4%), had a median size of 3.1 cm (range, 0.5–21 cm), and invaded vessels in 33.5%. Almost all patients (97.8%) had good liver function graded as Child Pugh grade A. Using AJCC T-staging systems, 58.4% of the patients were classified as T1a and 1b, followed by 30%, 5%, and 6.6% as T2, 3 and 4, respectively. Based on imaging findings, most patients (82.8%) were staged as BCLC 0 and A, followed by 11.1% and 6.1% staged as BCLC B and C, respectively. On pathologic examination, 74 (9.6%) showed the gross morphology of the infiltrative type HCC, while on imaging findings, infiltrative type was observed in 69 patients. The remaining patients showed nodular type HCCs.

### 3.2. Clinical Characteristics According to the Gross Morphology of HCC: Infiltrative vs. Nodular Type

The baseline clinical characteristics of patients with infiltrative and nodular HCC types were analyzed (Table 1). Patients with infiltrative HCCs showed a higher proportion of increased AFP level >20 ng/mL (71.2% vs. 46.3%, *p* < 0.001) and PIVKA-II level >50 mAU/mL (80.7% vs. 50.3%, *p* < 0.001). They also demonstrated a higher incidence of multifocal (≥2) tumors (35.1% vs 20.1%, *p* = 0.003), tumor size ≥ 4 cm (75.7% vs. 34.3%, *p* < 0.001), micro and major vessel invasion (45.9% vs. 26.4% and 23% vs. 3.3%, respectively, *p* < 0.001) than those with nodular type HCC. Therefore, advanced T-stage (AJCC-T ≥ 3) tumors were more frequently observed in HCCs of the infiltrative type than in the nodular type (36.5% vs. 9%, *p* < 0.001). Similarly, many patients with infiltrative type HCC (42%) were staged as BCLC stage B and C, whereas only 14.8% of those with the nodular type were staged as B and C (*p* < 0.001). Clinical and laboratory findings did not differ significantly between the two groups, except for a subtly higher median bilirubin level in patients with infiltrative HCC than in those with nodular HCC (0.68 vs. 0.80 mg/dL, *p* = 0.01).

### 3.3. Tumor Factors Associated with OverAll Survival

As of December 2018, 260 (33.6%) patients died with a higher incidence of mortality in those with infiltrative type HCC than in those with the nodular type (64.9% vs. 30.3%, *p* < 0.001; Figure 2a). In the multivariate analyses of all patients, tumor factors such as larger tumor size (≥4 cm), multifocality, micro or major vessel invasion, and gross morphology of the infiltrative type, along with well-known clinical markers of hepatic function such as albumin and prothrombin time, were independently associated with worse OS in all HCC-resected patients (all, *p* < 0.05) (Table 2).

Next, subgroup survival analysis was performed based on tumor size since it was an important factor in predicting OS. We found that the survival difference between the patients with infiltrative type HCC and those with the nodular type was remarkable in those with larger tumors (≥4 cm) (log rank, *p* < 0.001), but not in those with smaller tumors (<4 cm) (*p* = 0.103), as shown in Figure 2b,c. Multivariate analyses of subgroups also demonstrated that infiltrative HCC had an independent association with worse OS only in patients with large tumors ≥4 cm, but not in those with tumors <4 cm, even after adjusting for other factors (Table 2).

### 3.4. Impact of Large Infiltrative Type HCC on Current Staging Systems

Since the infiltrative HCCs that were larger than 4cm (large infiltrative type HCC) had a significant impact on survival, we assessed whether the current staging systems can stratify patients according to survival in those with this subtype of HCC. When the AJCC 8th T-staging system was applied to 56 patients with large infiltrative HCCs, the current AJCC staging algorithm for HCC was unable to discriminate the risk of death according to T-stages (Figure 3a).

Furthermore, among 286 patients who were classified as AJCC stage T1b, 14 patients (4.9%) had large infiltrative type HCCs, and they showed a significantly lower survival rate than those with other gross morphologic types within the same T-stage (log rank, *p* < 0.001). Similarly, 17 (7.3%) out of 232 AJCC T2-classified HCC patients had large infiltrative type HCC, and showed a remarkably lower survival rate than those with other subtypes within the same T-stage (*p* < 0.001) (Figure 3b,c). However, this deteriorative impact of large infiltrative type HCC was not seen in advanced T-stages (T3 or T4) (Figure 3d,e).

When BCLC staging was applied to the patients with large infiltrative type HCC based on the imaging findings, the staging system was unable to precisely discriminate the prognosis of these patients (Appendix A). Of 475 patients staged as BCLC A, 30 (6.3%) with large infiltrative HCC, and 9 out 86 (10.5%) patients in BCLC B with large infiltrative HCC, showed a markedly lower survival rate than those with other subtypes within the same sub-stage. (Appendix A). Similarly, as in AJCC T3 or T4, the adverse impact of this subtype of HCC was not significant in the BCLC stage C (Appendix A).

### 3.5. The Effect of the Modification of Staging Rule for Large Infiltrative HCCs

Considering the obviously deteriorative effect on the survival of patients with large infiltrative HCCs, especially for the cases staged as early stages in the current AJCC-T or BCLC staging criteria, we attempted to modify the staging rule for the large infiltrative type by reassigning them to more advanced stages such as AJCC-T3 or -T4 and BCLC B or C, and then evaluating the improved effect of those modifications on staging performance.

When all the large infiltrative HCCs classified as AJCC-T1b or T2 by the current criteria were shifted to T3, these T3-modified AJCC-T staging criteria significantly reduced the AIC value compared with that of the current criteria (3090.2 vs. 3108.9) while the c-index was increased (0.69 vs. 0.67, *p* < 0.001) (Table 3). The improved discriminative ability was clearly demonstrated by a more distinct difference in the survival between the T2 and T3 stages in the modified criteria (Figure 4a,b). Next, we repeated the analysis after reassigning all the large infiltrative HCCs to T4, and it was found that T4-modified AJCC-T staging criteria also had a better performance than the current AJCC-T criteria with the AIC value of 3088.8 and a c-index of 0.69 (*p* < 0.001) (Table 3) (Figure 4c), although there was no further improvement when compared to the T3-modified AJCC-T staging criteria.

In addition, another analysis was performed to observe if there was an improvement in the performance of BCLC staging after reassigning the large infiltrative type HCC to more advanced stages. The large infiltrative HCCs staged as BCLC A by the current criteria were reassigned to BCLC B, while all large infiltrative HCCs staged as BCLC B were reassigned to BCLC C. Again, these modified BCLC staging had lower AIC values (3069.7 vs. 3088.8) and higher C-indices (0.69 vs 0.67, *p* < 0.001) than those of the current BCLC staging criteria (Table 3).

## 4. Discussion

To date, this is the first study that enrolled all HCC patients who had undergone hepatectomy, conducted a pathologic review to identify the presence of infiltrative morphology in every single resected specimen, and documented its poor prognosis. Although the occurrence of infiltrative HCC is less frequent than nodular or multinodular confluent HCC, it is not rare as this morphologic subtype accounted for 7 to 13% of HCC cases in previous studies and 9.6% in our study [20,21]. The classic cross-sectional characteristics of HCC can be applied to the nodular type HCC but typical image findings of the infiltrative type HCC are poorly documented in the literature and its permeative appearance on cross-sectional imaging makes it difficult to be counted [22,23]. To overcome this limitation, all resected HCC specimens in this study were analyzed by experienced pathologists to confirm the gross morphology and presence of microvascular invasion.

In the present study, we validated tumor gross morphology, as well as the notorious factors of tumor burden (larger tumor size, multifocal tumors, and vessel invasion), and liver function were independently associated with patient survival. Patients with the infiltrative type HCCs demonstrated a remarkably worse OS than those with the nodular type. This could be explained by the larger tumor burdens seen already at the time of diagnosis in patients with infiltrative type HCC as represented by a higher incidence of larger, multifocal, vessel-invading tumors and a higher serum level of AFP and PIVKA-II compared to those with the nodular type HCC. In addition, the functional reserve of the liver (serum albumin and albumin levels) was slightly inferior in patients with the infiltrative type HCC. Nevertheless, the independent ill effect on survival of the infiltrative type HCC was evident even after adjusting well-known prognostic factors for HCC. This result is in accordance with previous studies reporting the poor prognostic impact associated with the infiltrative type HCC [9,12,23,24]. Therefore, we believe that the infiltrative type gross morphology seen in some HCCs represents their aggressive biological behavior at a certain time point in addition to the tumor burden, and consequently, it could be an objective prognostic marker that helps to stage HCCs.

However, we also found that the prognostic significance of the infiltrative type HCC differed depending on the tumor size. Subgroup analysis according to the tumor size revealed that the infiltrative type of gross morphology had an independent role in predicting death in conjunction with the current AJCC T-staging and BCLC staging systems only when the tumor was 4 cm or larger, but not in tumors <4 cm. The 5-year OS rate of patients with tumor size ≥4 cm was 61.1% for the nodular type HCC, while it decreased to 28.3% for the infiltrative type, but there was no significant difference in survival between the two gross types in tumors <4 cm, indicating that the gross morphology of tumors is an important factor that should be considered when staging HCCs, especially for larger tumors.

Presumably, this differential impact on survival according to the size of the infiltrative type HCC might be explained by the following reasons. First, as tumors grow larger, the risk of occult spread beyond the border of resection is higher because of its infiltrative growth. Second, the increase in tumor volume might raise the likelihood of inaccuracy in pathologic examination so that the aggressiveness of infiltrating HCCs cannot be fully defined solely on the pathologic parameters such as multifocality or vascular invasion, which are used for conventional staging. Third, the irregular and poorly demarcated boundary of the infiltrative type HCC makes it difficult to define its true number at staging.

In real clinical practice, however, enough attention has not been given to large infiltrative type HCCs when staging them, thereby leading to a significant proportion of infiltrative HCCs being classified at better stages, as observed in our study. For instance, according to the current AJCC-T staging criteria, 25% of the large infiltrative type HCCs were classified to T1b, 30.4% to T2, 12.5% to T3 and 32.1% to T4 in the present study. The problem is that the patients with the large infiltrative type HCC assigned to T1b and T2 had a significantly lower survival rate than other patients in the same stages, unlike in the T3 and T4 stages. T1b stage that comprises a solitary tumor >2 cm without vascular invasion is known to be associated with good prognosis. In the present study, however, it only applies to the nodular type or small HCCs <4 cm, but not the large infiltrative type. Similarly, the large infiltrative HCCs that were classified as T2 stage in the current criteria, which included solitary tumors >2 cm with vascular invasion or multifocal tumors ≤5 cm, had a significantly worse impact on survival than the other subtypes of HCCs. In contrast, the large infiltrative type of HCCs that were assigned to T3 or T4 appear to have no staging inaccuracy as these stages basically included the cases with poor prognostic factors such as large multiple tumors and the presence of major vessel invasions, respectively. The same problem was encountered in patients with the large infiltrative type HCC assigned to BCLC stage A and B by the current criteria.

Based on the above results, we found that HCCs of the large infiltrative type should be assigned to the advanced stages beyond T1 or T2 of the AJCC staging or beyond BCLC stage A or B. Therefore, we recommend assuming the large unifocal infiltrative type HCCs on surgical specimen as tumors with multiple foci and reassigning them from AJCC-T1 and T2 to AJCC-T3, or assuming all large infiltrative HCCs staged AJCC-T1 to T3 as those with macrovascular invasion and reassigning them to AJCC-T4. Second, for BCLC staging, we recommend any large unifocal-looking infiltrative type HCCs staged BCLC-A on imaging studies to be reassigned to BCLC-B, while definitely multifocal HCCs initially staged BCLC-B to BCLC-C. Indeed by doing so, we were able to substantially increase the prognostic performance of the current AJCC-T staging or BCLC staging as suggested by the significant changes in AIC values and c-indices.

Briefly, in the present study, we fully investigated the clinical features of HCCs according to their gross morphology in patients who underwent liver resection, and clarified the prognostic importance of the infiltrative type HCC. We found that the large infiltrative type HCC ≥4 cm was independently associated with a very poor prognosis, even in conjunction with the current AJCC T-staging and BCLC staging systems. Next, we addressed the concern on the AJCC 8th T-staging and BCLC staging system for this subtype for the first time, and proposed a necessity of modification of current staging criteria by reassigning large infiltrative HCCs to at least AJCC-T3, while one-stage-up for BCLC-A and BCLC-B. This modification not only allowed better prognostication, but also made the staging system practically feasible. However, a limitation of this study is its retrospective nature, and hence, the findings should be validated in larger cohorts.

## 5. Conclusions

In conclusion, we propose that large infiltrative type HCCs should be assigned to more advanced stages in current staging systems in consideration of their independent prognostic impact. This will enable finer stratification of HCC patients and provide more accurate prognostic competence.

## Figures and Tables

**Figure 1 cancers-12-02589-f001:**
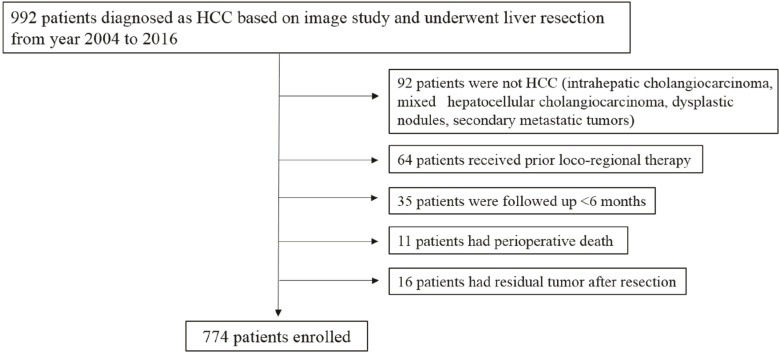
Flow chart of patient selection after liver resection.

**Figure 2 cancers-12-02589-f002:**
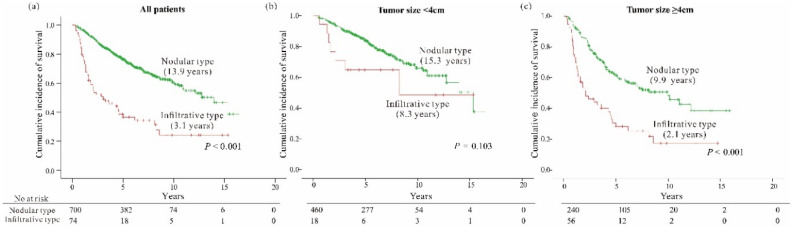
Overall survival according to tumor morphology for (**a**) all patients, (**b**) tumor size <4 cm and (**c**) tumor size ≥4 cm. Median survival time is shown in brackets.

**Figure 3 cancers-12-02589-f003:**
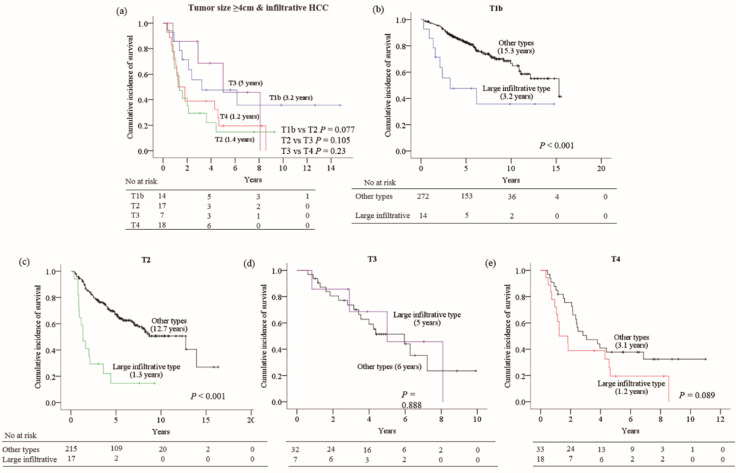
Overall survival rate according to (**a**) current T-staging system for large infiltrative hepatocellular carcinoma (HCC). Impact of large infiltrative HCC on different stages, (**b**) T1b, (**c**) T2, (**d**) T3 and (**e**) T4. Median survival time is shown in brackets.

**Figure 4 cancers-12-02589-f004:**
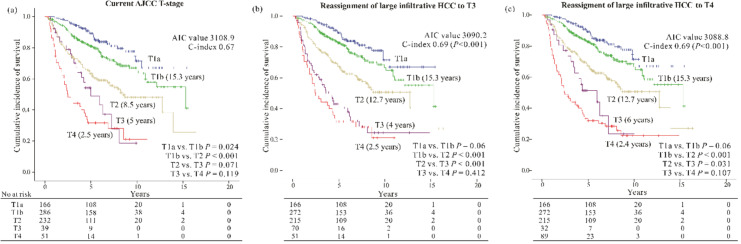
Overall survival according to (**a**) current T-staging system. Modified T-staging system after reassigning large infiltrative HCC to (**b**) T3 and (**c**) T4. Median survival time is shown in brackets.

**Table 1 cancers-12-02589-t001:** Baseline characteristics of patients according to the gross morphology of hepatocellular carcinomas.

Variables		Total	Infiltrative Type	Nodular Types	*p*-Value
Number of patients		774	74 (9.6)	700 (90.4)	
Clinical factors					
Age >60 years, *n* (%)		312 (40.3)	28 (37.8)	284 (40.6)	0.648
Male, *n* (%)		607 (78.4)	57 (77)	550 (78.6)	0.759
Etiologies					
	HBV	601 (77.6)	58 (78.4)	543 (77.6)	0.138
	HCV	48 (6.2)	1 (1.4)	47 (6.7)	
	Alcoholic	67 (8.7)	6 (8.1)	61 (8.7)	
	Cryptogenic	58 (7.5)	9 (12.2)	49 (7)	
Laboratory findings					
	ALT, IU/L	33 (6–413)	38 (6–166)	33 (6–413)	0.244
	AST, IU/L	36 (9–374)	38 (16–318)	35 (9–374)	0.34
	Albumin, g/dL	4.2 (2.8–5.1)	4.1 (3.1–5.1)	4.2 (2.8–5.0)	0.069
	Total bilirubin, mg/dL	0.69 (0.1–9.8)	0.80 (0.3–9.78)	0.68 (0.1–3.69)	0.01
	PT INR	1.04 (0.71–1.78)	1.04 (0.76–1.4)	1.04 (0.71–1.78)	0.76
	AFP >20 ng/mL (*n* = 752)	367 (48.7)	52 (71.2)	315 (46.3)	<0.001
	PIVKA-II >50 mAU/mL (*n* = 606)	322 (53.1)	46 (80.7)	276 (50.3)	<0.001
Child-Pugh grade					
	A	757 (97.8)	71 (95.9)	686 (98)	0.252
	B	17 (2.2)	3 (4.1)	14 (2)	
Tumor factors					
	No. of tumors, *n* (%)				
	Unifocal	607 (78.4)	48 (64.9)	559 (79.9)	0.003
	Multifocal	167 (21.6)	26 (35.1)	141 (20.1)	
Tumor size, *n* (%)					
	<4 cm	478 (61.8)	18 (24.3)	460 (65.7)	<0.001
	≥4 cm	296 (38.2)	56 (75.7)	240 (34.3)	
Vessel invasion, *n* (%)					
	None	515 (66.5)	23 (31.1)	492 (70.3)	<0.001
	Micro	219 (28.3)	34 (45.9)	185 (26.4)	
	Major	40 (5.2)	17 (23)	23 (3.3)	
AJCC 8th T-stage, *n* (%)					
	1a	166 (21.4)	2 (2.7)	164 (23.4)	<0.001
	1b	286 (37)	17 (23)	269 (38.4)	
	2	232 (30)	28 (37.8)	204 (29.1)	
	3	39 (5)	7 (9.5)	32 (4.6)	
	4	51 (6.6)	20 (27)	31 (4.4)	
BCLC, *n* (%)					
(Number of patients)		774	69 (8.9)	705 (91.1)	
	0	166 (21.4)	0	166 (23.5)	<0.001
	A	475 (61.4)	40 (58)	435 (61.7)	
	B	86 (11.1)	10 (14.5)	76 (10.8)	
	C	47 (6.1)	19 (27.5)	28 (4)	

Tumor factors and AJCC 8th T-staging were based on pathologic findings, while BCLC staging was based on imaging findings. AFP, α-fetoprotein; AJCC, American Joint Committee on Cancer; ALT, alanine aminotransferase; AST, aspartate aminotransferase; BCLC, Barcelona Clinic Liver Cancer; HBV, hepatitis B virus; HCV, hepatitis C virus; PT INR, prothrombin time international normalized ratio; PIVKA-II, protein induced by vitamin K absence or antagonist-II. Continuous variables are presented as median (range) and categorical variables as number of cases (%).

**Table 2 cancers-12-02589-t002:** Factors associated with overall survival.

Variables			Univariate Analysis		Multivariate Analysis					
			HR (95% CI)	*p*	Model 1		Model 2		Model 3	
					HR (95% CI)	*p*	HR (95% CI)	*p*	HR (95% CI)	*p*
**All patients**										
Clinical factors										
	Age >60 years		1.043 (0.812–1.339)	0.742						
	Male		1.485 (1.067–2.067)	0.019	1.358 (0.973–1.896)	0.072				
	Etiologies									
	HBV		–	0.023						
	HCV		1.303 (0.792–2.142)	0.297		0.507				
	Alcoholic		1.733 (1.184–2.536)	0.005						
	Cryptogenic		1.37 (0.888–2.113)	0.154						
Laboratory findings										
	Albumin >4 g/dL		0.513 (0.402–0.655)	<0.001	0.588 (0.458–0.756)	<0.001				
	Total bilirubin >1 mg/dL		1.153 (0.85–1.563)	0.36						
	INR >1		1.286 (0.977–1.692)	0.073	1.319 (0.996–1.748)	0.050				
Tumor factors										
	Tumor size, ≥4 cm		2.545 (1.992–3.251)	<0.001	1.836 (1.405–2.399)	<0.001				
	Infiltrative vs. nodular types		3.28 (2.394–4.494)	<0.001	2.035 (1.456–2.845)	<0.001				
	Multifocal vs unifocal		2.164 (1.661–2.819)	<0.001	1.503 (1.133–1.994)	0.005				
	Presence of vessel invasion									
		None	–	<0.001		<0.001				
		Micro	2.159 (1.662–2.804)	<0.001	1.669 (1.267–2.198)	<0.001				
		Major	5.686 (3.833–8.434)	<0.001	2.372 (1.523–3.693)	<0.001				
**Tumor size <4 cm group**										
Clinical factors										
	Male		1.75 (1.032–2.967)	0.038	1.629 (0.955–2.778)	0.073	1.652 (0.97–2.812)	0.065	1.602 (0.94–2.729)	0.083
Laboratory findings										
	Albumin >4 g/dL		0.513 (0.355–0.74)	<0.001	0.597 (0.409–0.872)	0.008	0.597 (0.409–0.873)	0.008	0.585 (0.401–0.855)	0.006
	PT INR >1		1.96 (1.21–3.173)	0.006	1.7 (1.04–2.779)	0.034	1.752 (1.07–2.869)	0.026	1.791 (1.093–2.935)	0.021
Tumor factors										
	Tumor size, >2 cm		1.795 (1.199–2.686)	0.004	1.49 (0.983–2.257)	0.06				
	Infiltrative type vs. others		1.879 (0.87–4.06)	0.109						
	Multifocal vs unifocal		1.561 (0.988–2.465)	0.056		0.176				
	Presence of vessel invasion									
		None	-	<0.001		0.004				
		Micro	1.461 (0.956–2.231)	0.079	1.399 (0.91–2.151)	0.126				
		Major	7.031 (3.048–16.22)	<0.001	3.995 (1.688–9.458)	0.002				
AJCC 8th T-stage										
	T1a		-	<0.001			-	<0.001		
	T1b		1.425 (0.887–2.289)	0.143			1.274 (0.79–2.056)	0.321		
	T2		2.036 (1.262–3.285)	0.004			1.968 (1.217–3.182)	0.006		
	T4		9.17 (3.782–22.234)	<0.001			5.613 (2.26–13.941)	<0.001		
BCLC stage										
	0		–	<0.001					-	<0.001
	A		1.52 (0.985–2.344)	0.058					1.39 (0.899–2.151)	0.139
	B		3.959 (2.016–7.772)	<0.001					4.134 (2.096–8.156)	<0.001
	C		9.199 (3.794-22.304)	<0.001					5.597 (2.255–13.894)	<0.001
**Tumor size ≥4 cm group**										
Laboratory findings										
	Albumin >4 g/dL		0.543 (0.39–0.755)	<0.001	0.6 (0.429–0.838)	0.003	0.592 (0.423–0.828)	0.002	0.581 (0.416–0.811)	0.001
Tumor factors										
	Size >5 cm		1.598 (1.109–2.303)	0.012		0.614				
	Infiltrative type vs. others		2.581 (1.793–3.715)	<0.001	2.181 (1.496–3.181)	<0.001	2.318 (1.585–3.39)	<0.001	2.301 (1.578–3.357)	<0.001
	Multifocal vs unifocal		1.978 (1.408–2.779)	<0.001	1.517 (1.063–2.165)	0.022				
	Presence of vessel invasion									
		None		<0.001		0.001				
		Micro	2.098 (1.448–3.038)	<0.001	1.838 (1.258–2.686)	0.002				
		Major	3.655 (2.245–5.951)	<0.001	2.271 (1.345–3.834)	0.002				
AJCC 8th T-stage										
	T1b		–	<0.001				<0.001		
	T2		2.43 (1.58-3.738)	<0.001			2.442 (1.588–3.756)	<0.001		
	T3		2.348 (1.35–4.084)	0.003			2.014 (1.155–3.512)	0.014		
	T4		3.153 (2.13–5.796)	<0.001			2.476 (1.473–4.16)	0.001		
BCLC stage										
	A		–	<0.001						<0.001
	B		1.913 (1.288–2.842)	0.001					2.115 (1.415–3.161)	<0.001
	C		3.571 (2.341–5.448)	<0.001					2.753 (1.786–4.242)	<0.001

Model 1, multivariate Cox models including clinical factors, laboratory findings and tumor factors based on pathologic findings. Model 2, multivariate Cox models including clinical factors, laboratory findings and AJCC T-stage based on pathologic findings. Model 3, multivariate Cox models including clinical factors, laboratory findings and BCLC stage based on image findings. AFP, α-fetoprotein; AJCC, American Joint Committee on Cancer; ALT, alanine aminotransferase; AST, aspartate aminotransferase; BCLC, Barcelona Clinic Liver Cancer; HR, hazard ratio; PT INR, prothrombin time international normalized ratio; PIVKA, protein induced by vitamin K absence or antagonist II.

**Table 3 cancers-12-02589-t003:** Performance of AJCC 8th T-stage and BCLC stage before and after reassignment of large infiltrative hepatocellular carcinomas.

Staging System	AIC	C-Index	LHRχ^2^
Original AJCC 8th T-stage	3108.946	0.673 (0.642–0.704)	87.73 (<0.001)
T3 modified T-stage	3090.167	0.692 (0.661–0.724) *	106.52 (<0.001)
T4 modified T-stage	3088.784	0.693 (0.662–0.725) *	107.89 (<0.001)
Original BCLC stage	3088.764	0.667 (0.637–0.696)	105.82 (<0.001)
Modified BCLC stage	3069.662	0.686 (0.656–0.716) *	125.02 (<0.001)

* *p*-value < 0.001.

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
