# Peer review of "How Should We Assign Large Infiltrative Hepatocellular Carcinomas for Staging?"

_cancers, 2020, doi:10.3390/cancers12092589_

Round 1

Reviewer 1 Report

Interesting work, especially when reclassification was attempted. I have some observations:

1) Diagnosis was made applying AASLD criteria, the citation referred to 2018 but data are collected since 2004. Please fix this discrepancy.

2) Please unequivocally describe that tumor features are based on histological findings. Add a footnote in the table 1 about this issue. In this regard, BCLC is not based on pathology but on radiology. Thus, the inclusion of this system does not seems adequate. In addition, to be strictly adherent to BCLC, all patients were resected, thus, had stage migration from a more advanced stage to stage A (this aspect of bclc is embarrassing). This underlines that in the present manuscript, Authors should show the radiological BCLC criteria (and the consequent aderhence) and not histological BCLC which did not exist. All the subsequent analyses including BCLC should be removed.

3) Please clearly describe whether the Akaike information criterion (AIC) calculated derived from the simple inclusion of each staging system as an unique variable (so that k= number of variables in the model = 1) or considered that each staging system was formed by more than 1 variable (i.e. AJCC = T + N + M = 3 variables, k =3).

4) Figure 2 can be summarized in a Table as well as most of supplementary survival curves. Authors can provide medians, or 1-y, 3-y, 5-y survival rates and log-rank values, as it is more comfortable for them.

5) Multivariable analysis is really cryptic. Three orders of p-values are reported with an inexplicable footnote. In addition, HR for multivariable analysis must be provided.

Author Response

Interesting work, especially when reclassification was attempted. I have some observations:

1) Diagnosis was made applying AASLD criteria, the citation referred to 2018 but data are collected since 2004. Please fix this discrepancy.

  • Thank you for the comment. We have made changes under “Imaging findings” according to guideline that was published before and after 2011. The changes are shown below and are highlighted in the manuscript.
  • The preoperative diagnosis of HCC was based on imaging criteria using CT and MRI with contrast agents. According to the EASL or AASLD guidelines, nodules detected before 2011 were diagnosed as HCC if they had typical appearance of HCC in two and one imaging studies for nodules between 1-2cm and larger than 2cm, respectively [13,14]. However, after 2011, one dynamic imaging study with typical findings was sufficient to diagnose HCC for nodules larger than 1cm [15].

2) Please unequivocally describe that tumor features are based on histological findings. Add a footnote in the table 1 about this issue. In this regard, BCLC is not based on pathology but on radiology. Thus, the inclusion of this system does not seems adequate. In addition, to be strictly adherent to BCLC, all patients were resected, thus, had stage migration from a more advanced stage to stage A (this aspect of bclc is embarrassing). This underlines that in the present manuscript, Authors should show the radiological BCLC criteria (and the consequent aderhence) and not histological BCLC which did not exist. All the subsequent analyses including BCLC should be removed.

  • Thank you for the comment. Any findings that were based on histological findings has been noted in table 1 and 2 as footnote.
  • We appreciate your critical comment. BCLC staging is based on radiologic findings and is inappropriate to classify according to histologic findings.
  • Therefore, we have gone through the image findings and presence of infiltrative pattern was reanalyzed using CT or MRI. Based on the imaging finding, 69 patients had infiltrative type HCC. Prognostic efficacy of BCLC was reanalyzed.
  • The representative images are shown in supplementary figure 2 and any results regarding BCLC are all based on imaging findings.

3) Please clearly describe whether the Akaike information criterion (AIC) calculated derived from the simple inclusion of each staging system as an unique variable (so that k= number of variables in the model = 1) or considered that each staging system was formed by more than 1 variable (i.e. AJCC = T + N + M = 3 variables, k =3).

  • Thank you for the comment. Akaike information criterion (AIC) calculated calculated based on the Cox proportional hazards regression model. We fitted the regression model for each staging system separately and included the stages as categorical variable
  • For AJCC only T-staging system was evaluated and therefore the stages such as Ia, Ib, II, III and IV are included as categorical variable and applies the same to BCLC stages.

4) Figure 2 can be summarized in a Table as well as most of supplementary survival curves. Authors can provide medians, or 1-y, 3-y, 5-y survival rates and log-rank values, as it is more comfortable for them.

  • Thank you for the comment. Median survival time has been added in every figures.
  • In addition we have also added number at risk for KM survival plot.

5) Multivariable analysis is really cryptic. Three orders of p-values are reported with an inexplicable footnote. In addition, HR for multivariable analysis must be provided.

  • Thank you for the comment. We have reported three orders p-values and added HR for multivariable analysis.
  • Table has been reorganized to be more explicable especially table 2.

Reviewer 2 Report

The manuscript entitled "How Should We Assign Large Infiltrative
3 Hepatocellular Carcinomas for Staging?" retrospectively found the prgnostic significance of different type of HCC imaging patterns, that is infiltrative versus nodular. Only one major issues should be addressed.

Major issue

The imaging diagnosis criterion of both infiltrative and nodular types of HCC should be clearly defined in the material and method section. In addition, should different imaging definition of infiltrative and nodular types of HCC affects the prognosis should also be discussed.

Author Response

The manuscript entitled "How Should We Assign Large Infiltrative Hepatocellular Carcinomas for Staging?" retrospectively found the prognostic significance of different type of HCC imaging patterns, that is infiltrative versus nodular. Only one major issues should be addressed.

Major issue

The imaging diagnosis criterion of both infiltrative and nodular types of HCC should be clearly defined in the material and method section. In addition, should different imaging definition of infiltrative and nodular types of HCC affects the prognosis should also be discussed.

  • Thank you for the critical comment. We absolutely agree that imaging diagnosis criteria are required for both nodular and infiltrative types of HCC especially for BCLC staging which is based on imaging.
  • Therefore, we have made clear definition of both types of HCCs in “Imaging findings” under method section and the changes are as follows.
  • Nodular HCC was defined as tumors with well-circumscribed borders showing typical radiologic characteristics of HCC with early arterial hyperenhancement and washout pattern on portal venous phase images. Infiltrative HCC was defined when tumors presented poorly demarcated indistinct margins with diffuse permeative appearance, demonstrating very inhomogeneous or miliary enhancement pattern on arterial phase images and corresponding washout on more delayed phase images (Supplementary Fig. 1).
  • The most representative imaging and pathologic findings of nodular and infiltrative type HCCs are shown in Supplementary figure 1.
  • The prognosis of infiltrative type HCC based on imaging finding consistently showed worse prognosis than the nodular type HCC obviously in BCLC A and B. The results are shown in Supplementary figure 2 and the changes made are highlighted in the manuscript.

Round 2

Reviewer 1 Report

Authors clarified some aspects from the observation #2, and I thank them for their effort. It remains an important aspect to define, that is, how they classified according to BCLC. I suppose that a single nodule <2cm was BCLC-0, a tumor within Milan criteria but not fulfilling BCLC-0 was a BCLC-A. Now, how was defined a single tumor >5cm? They were resected, thus, to be strictly adherent to BCLC it should be a BCLC-A. Somewhere they must define BCLC, the aspect now highlighted was the basis of my previous suggestion to remove BCLC from the manuscript. Otherwise they can refer to some of these: J Hepatol. 2017 Jul;67(1):173-183; Ann Surg. 2013 May;257(5):929-37; Hepatology. 2015 Mar;61(3):905-14.

Authors did not understand my observation #3. Fortunately, they did not compare different staging systems rather the original with the modified ones. AJCC was considered only in its T component, so that the problem did not exist. For BCLC they luckly incurred in the same denominator so that the final comparison was not biased.

Author Response

Authors clarified some aspects from the observation #2, and I thank them for their effort. It remains an important aspect to define, that is, how they classified according to BCLC. I suppose that a single nodule <2cm was BCLC-0, a tumor within Milan criteria but not fulfilling BCLC-0 was a BCLC-A. Now, how was defined a single tumor >5cm? They were resected, thus, to be strictly adherent to BCLC it should be a BCLC-A. Somewhere they must define BCLC, the aspect now highlighted was the basis of my previous suggestion to remove BCLC from the manuscript. Otherwise they can refer to some of these: J Hepatol. 2017 Jul;67(1):173-183; Ann Surg. 2013 May;257(5):929-37; Hepatology. 2015 Mar;61(3):905-14.

→Thank you for your comment and your request for the clarification of BCLC classification.

→We have gone through the radiologic findings not histologic findings for BCLC staging. Therefore, the BCLC staging are entirely based on imaging findings strictly following BCLC staging system. As the reviewer has mentioned any single tumor larger >5cm without gross vascular invasion in imaging finding is classified as BCLC A.

→We appreciate your suggestion and to clarify that any single tumor without gross vascular invasion, disregarding the tumor size was classified as BCLC A according to original BCLC original staging system, we have added this information and highlighted under the “Staging System”.

Authors did not understand my observation #3. Fortunately, they did not compare different staging systems rather the original with the modified ones. AJCC was considered only in its T component, so that the problem did not exist. For BCLC they luckly incurred in the same denominator so that the final comparison was not biased.

→Thank you again for your comment. As the reviewer mentioned we did not compare between different staging systems but rather within the same staging system, comparing between the original with the modified ones. Once again we appreciate your critical observation. 

Reviewer 2 Report

Thanks for author's effort to refine the manuscript. The study is helpful to make up for short comings of current staging systems.

Author Response

Thank you for your supportive comments.